# The Unique Morphology of Coconut Petiole Fibers Facilitates the Fabrication of Plant Composites with High Impact Performance

**DOI:** 10.3390/polym15092200

**Published:** 2023-05-05

**Authors:** Shiqiang Fu, Hongwu Wu, Kang Zhu, Zhouxiang Zhao, Zhifang Liang

**Affiliations:** The Key Laboratory of Polymer Processing Engineering of Ministry of Education, National, Engineering Research Center of Novel Equipment for Polymer Processing, Guangdong, Provincial Key Laboratory of Technique and Equipment for Macromolecular Advanced, Manufacturing, South China University of Technology, Guangzhou 510640, China

**Keywords:** bio–composite, coconut petiole fiber, polylactic acid, alkali treatment, mechanical property, interfacial adhesion

## Abstract

The present work explored alkali–treated coconut petiole fibers (ACPFs) characterization and the effect of fiber loadings on the mechanical properties of poly (lactic acid) (PLA)/ACPF composites for the first time. The physical, mechanical, and interfacial properties, as well as the morphology of the ACPFs were reported. It was found that ACPFs with a density of 0.92 g/cm^3^ have average tensile strength and tensile modulus equal to 355.77 MPa and 5212.36 MPa. The interfacial strength between ACPFs and PLA was high (14.06 MPa), attributed to the micro–sized holes on the fibers, as established from SEM micrographs. Then composites with varying fiber loadings were fabricated by melt–blending and compression molding. The mechanical (tensile, flexural, and impact) performance of composites was reported. Based on the high interfacial strength between fibers and PLA and the unique “spiral” structure of fibers, the composites reached a high impact strength of 8.2 kJ/m^2^ and flexural modulus of 6959.70 MPa at 50 wt.%, representing 150% and 50% improvement relative to pure PLA.

## 1. Introduction

Nowadays, oil resources are gradually becoming scarce, and the environmental consciousness of people is gradually increasing. People tend to use renewable materials [1,2,3,4,5]. Among various environmentally friendly materials, natural fibers reinforced polymer composites are getting more attention because of their degradability and low cost [6,7,8,9]. Nowadays, natural fiber–reinforced polymer composites are replacing synthetic counterparts in a variety of locations such as the automotive field and construction industries.

Plant fibers are composed of cellulose, hemicellulose, lignin, waxes, and pectin. The climatic conditions, soil, and part of the plant decide the ratio of its constituents, which determines the mechanical properties of the fibers [10,11]. The most used natural fibers are kenaf, jute, sisal, and flax [12,13,14]. Before the manufacture of composites, the surface of fibers was usually modified to enhance the bond between the fibers and the matrix [15,16,17]. The surface treatments include alkali, silane, peroxide, benzoyl chloride treatment, etc. Alkali treatment was often used because of its simple operation and good treatment effect [18,19,20]. Arthanarieswaran et al. reported that alkali treatment can partially remove impurities such as hemicellulose, lignin, and pectin from the fiber, and the cellulose content and crystallinity increased, which leads to a significant improvement in the mechanical properties of the fiber [21]. Kapatel reported that the fiber/matrix adhesion of jute/epoxy composites was improved by removing impurities from the jute fabric surface after alkali treatment, resulting in the improvement of the mechanical properties of the composites [22]. 

Coconut (*Cocos nucifera* L.) belongs to the palm family. The height of the plant is 15–20 m and its leaves are pinnate, 3–4 m long. The main producers are Sri Lanka, Malaysia, India, and the Philippines. Coconut is an abundant source of natural resources that can be used in many industrial fields. Coconut shells can be made into a variety of utensils and handicrafts. Fibers from different parts of the coconut palm tree can be extracted and utilized. Satyanarayana et al. examined the size, density, tensile strength, and percentage elongation of fibers extracted from the spathe, leaf sheath, and back of the petiole [23]. Xu et al. successfully extracted cellulose nanofibers from coconut palm petioles by grinding and chemical treatment. The fibers showed high crystallinity and good thermal stability and might be used to prepare fiber–reinforced materials [24]. Coconut petioles can also be used to make brushes and the fibers at the end of coconut petioles are exposed. It is therefore important to extract and characterize the fibers and use them as reinforced materials. PLA is a thermoplastic polymer produced from lactic acid with specific advantages such as good processability, low processing temperature, and good biocompatibility [25,26,27,28]. So far, much research has been conducted to use natural fibers to reinforce PLA [29,30,31]. Lv et al. successfully prepared sisal fiber/ PLA composites by melt–blending and compression molding. When the fiber content was 30 wt.%, the impact strength reached 4.5 kJ/m^2^. But few works of literature are available on composites fabricated with PLA reinforced by coconut petiole fibers [32].

The main aim and contribution of this study are to extract a new fiber from coconut petioles and investigate the reinforcing effect of its unique morphology on composites. The functional group and the morphology of alkali–treated coconut petiole fibers (ACPFs) were characterized with Fourier Transform Infrared Spectroscopy (FT–IR) and scanning electron microscope (SEM), respectively. In addition, density, tensile strength, tensile modulus, and interfacial strength were determined. Then PLA/ACPF composites were prepared by melt–blending and compression molding. The mechanical (tensile, flexural, and impact) properties of the PLA/ACPF composites with varying fiber loadings were studied. This work provides a basis for future research on this fiber as an alternative reinforcement in bio–composite materials.

## 2. Materials and Methods

### 2.1. Materials

PLA, 4032D, was obtained from Nature Works. Its density was 1.24 g/cm^3^, and its melt flow rate (MFR) was 7 g/10 min at 190 °C and 2.16 kg load. Coconut petioles were obtained from Zhanjiang, Guangdong Province. Sodium hydroxide (NaOH) reagent and acetone reagent were produced by Shandong Gushuo Biotechnology Co., Ltd (Jining, China). Water used in the experiments was deionized. 

### 2.2. Fiber Extraction and Pretreatment

Figure 1 shows the extraction process of coconut petiole fibers. First, coconut petioles were dried and knocked to separate coconut petiole fibers. Coconut petiole fibers were cut short and washed thoroughly in deionized water to eliminate unwanted impurities such as dust. The CPFs were dried in an oven for 8 h at 80 °C to remove water.

To remove impurities from the fiber surface, the fibers were alkali treated with 10 wt.% of NaOH solution for 2 h at 25 °C, which was beneficial for the fibers proved by Sullins et al. and Pickering et al. [33,34]. Then they were repeatedly washed with freshwater to remove NaOH until the pH of the washing water was 7. Then the fibers were dried in an oven for 8 h at 80 °C. The fibers during this phase were called alkali–treated coconut petiole fibers (ACPFs). 

### 2.3. Sample Preparation

ACPFs were cut into 2 cm of length and dried for 6 h at 80 °C with PLA. The water content of ACPFs after drying was 0.57%. An internal mixer (Poton 100, POTOP Experimental Analysis Instrument Co., Ltd., Guangzhou, China) was used to mix the ACPFs with PLA for 8 min at 180 °C with a rotor speed of 40 r/min. The rotor type was Roller, and the chamber volume was 55 cm^3^. First, PLA was added to the chamber. After the matrix was completely melted, the fibers were added. The fiber loadings changed from 0 to 50 wt.% with a step of 10 wt.%. After cooling down, the obtained mixtures were molded into sheets for tensile, flexural, and impact tests by a hot press (QLB–25D/Q, WUXI NO.1 Ltd., Wuxi, China). The temperature was 190 °C and the pressure was 10 MPa. Dumbbell–shaped specimens were prepared for tensile test and rectangular specimens were prepared for flexure and impact tests. Five samples were measured for each test, and the average values were recorded. 

### 2.4. Characterization of Fibers

#### 2.4.1. SEM and FT–IR Tests

The morphology of the cross–section and surface of CPFs and ACPFs was examined by a scanning electron microscope (Quanta 200, FEI Ltd., Eindhoven, The Netherlands). The surface was sprayed with gold for 3 min to improve its electrical conductivity. 

The FT–IR was identified using the FT–IR spectrometer (RHEOLOGIC5000, CEAST Ltd., Rome, Italy) from the range of 500 cm^−1^ to 4000 cm^−1^.

#### 2.4.2. Diameter of CPFs and ACPFs

The diameter of the CPFs and ACPFs was determined by a stereomicroscope (Stemi2000–C, ZEISS Ltd., Leipzig, Germany). Five samples were measured for each test, and the average values of the fibers were recorded.

#### 2.4.3. Density of CPFs and ACPFs

The pycnometer was used to test the density of fibers using the mass difference technique. The reagent used for the density test was liquid acetone with a density of 0.7845 g/cm^−3^. First, the fibers were put into the pycnometer, and the masses of the pycnometer before and after the placement were recorded as *m*_1_ and *m*_2_, respectively. A certain mass of acetone (*m*_3_) was added to the pycnometer. The final mass *m*_4_ was recorded and the density of the fibers could be calculated by the following equation: (1)$$\rho_{\mathrm{fiber}}=\left[\frac{m_{2}{-m}_{1}}{\left(m_{3}{-m}_{2}\right)\left(m_{4}{-m}_{2}\right)}\right]{*\rho }_{\mathrm{acetone}}$$

#### 2.4.4. Tensile Test and Microdroplet Debonding Experiment

The tensile test of the fibers was performed by a universal testing machine (Instron 5566, INSTRON Ltd., Norwood, MA, USA). The load applied on the fibers was 100 N, and the speed of the crosshead was 0.5 mm/min. Five samples were measured for each test and the average values were recorded.

The microdroplet debonding experiment was performed by the universal testing machine. PLA was melted into droplets and wrapped around one end of the fiber. The other end of the fiber passed through the fixture with holes. The fiber was able to pass through the fixture while the PLA droplets were not [35]. In Figure 2, it was observed that when tension was applied at one end of the fiber, the fiber could be pulled out of the droplets. The tensile rate was 0.5 mm/min. The interfacial shear strength (IFSS) *σ_i_* was calculated by the following equation:(2)$$\sigma_{i}=\frac{F_{i}}{{\pi \mathrm{dl}}_{a}}$$
where *F_i_* was the maximum load, *d* was the diameter of the fiber and *l_a_* was the embedded length of the fiber. *d* and *l_a_* were measured by a stereomicroscope (Stemi2000–C, ZEISS Ltd., Leipzig, Germany). Five samples were measured for each test and the average values were recorded.

#### 2.4.5. Thermogravimetric Analysis

Testing of thermal stability is essential for plant fibers used as reinforcing fillers. Thermogravimetric Analysis (TGA) was used to characterize and analyze the thermal stability properties of coconut petiole fibers before and after alkali treatment. The fibers were placed in a quartz crucible for thermogravimetric testing. The test was carried out under nitrogen protection at a purge rate of 20 mL/min. The heating rate was set to 10 °C/min and the temperature range was 30–600 °C during the test.

### 2.5. Composites Characterization

#### 2.5.1. SEM Test of Composites

The morphology of the impact–fractured sample surface was recorded using SEM (Quanta 200, FEI Ltd., Eindhoven, The Netherlands). The section was sprayed with gold for 3 min to improve its electrical conductivity. 

#### 2.5.2. Mechanical Properties of Composites

Tensile and flexural test. Tensile and flexural properties were conducted by a universal testing machine (Instron 5566, INSTRON Ltd., Norwood, MA, USA) following ISO 527–2:1993 and ISO 14125:1998. Tensile test specimens were placed in grips and then tested. The load applied to the specimens was 100 N, and the speed of the crosshead was 2 mm/min. Five samples were measured for each test and the average values were recorded.

Impact test. Impact testing was performed by ISO 180:2000 with an impact testing machine (PIT501B–2, WANCE Ltd., Shenzhen, China). Notches were prepared on specimens using a dedicated notching machine (GT–7016–A2, GOTECH Ltd., Hong Kong, China). The Izod impact samples were tested with a hammer capacity of 0.5 J. Five samples were measured for each test and the average values were recorded.

## 3. Results and Discussion

### 3.1. SEM Analysis

The surface morphology of CPFs and ACPFs is reported in Figure 3. CPFs were impregnated with a waxy external layer. This observation was typified in the investigation carried out by Ye et al. [36]. After alkali treatment, the waxy external layer was removed and the diameter of the fiber was smaller. The surface of ACPF was rougher with a large number of holes in micron–sized due to the removal of impurities from the fibers after alkali treatment, as illustrated in Figure 3b. This special porous structure was not found in known plant fibers such as sisal and coir fibers [37]. The dense holes facilitated the penetration of the resin into the fiber surface, thus improving the interfacial adhesion between the ACPFs and the matrix [38,39]. 

Figure 4 shows the sectional morphology of CPFs and ACPFs. The fibers were composed of a series of microfiber bundles as reported in Figure 4a,b. This structure of CPFs and ACPFs had similarities to that of sisal fibers in work completed by Lv et al. [32]. Compared to the CPFs, the binding components on the surface of the ACPFs as well as the binding components between the fiber bundles were removed and there was a slight collapse of the fiber bundles. The cross–section of the ACPFs was rough, and many grooves appeared, which was typified in the investigation carried out by Vijay et al. [40]. The longitudinal section morphology of the fibers might offer further information about the structure of the fibers. The microfibers presented an obvious spiral–pattern structure in Figure 4c,d. This special helical structure might have a significant impact on the mechanical properties of the fibers. When the fibers were pulled in tension, these microfibers might uncoil like springs with bending and twisting, thus giving the fibers a certain degree of tensile toughness. 

### 3.2. FT–IR Analysis

Figure 5 shows the FT–IR spectrum of CPFs and ACPFs. For CPFs, there were broad peaks at 3410 cm^−1^ and 1050 cm^−1^, and prominent peaks at 1736, 1630, 1380, and 1250 cm^−1^. According to Picard et al., the peak at 3410 cm^−1^ was characteristic of the stretching vibrations of O–H [41]. In addition, the peak at 1050 cm^−1^ was assigned to the C–O–H stretching in cellulose and hemicellulose. The peak at 1380 cm^−1^ represented the flexural vibration of C–H attributed to hemicellulose and cellulose. Furthermore, the peak at 1736 cm^−1^ represented the C=O stretching modes in hemicellulose and cellulose. The peak at 1736 cm^−1^ represented the conjugated C=O stretching vibrations in lignin. In addition, the peak at 1250 cm^−1^ represented the C–O stretching vibrations in lignin. Cellulose, hemicellulose, and lignin were confirmed to be present in CPFs by FT–IR analysis.

The distribution of the peaks in the FT–IR spectrum of ACPFs was similar to that of CPFs. However, the intensity of some peaks such as 1630 cm^−1^ and 1380 cm^−1^ decreased after alkali treatment. Some peaks disappeared such as 1736 cm^−1^ and 1250 cm^−1^. This indicated that parts of the pectin, hemicellulose, and lignin were removed by alkali treatment so that the pores on the surface of fibers were exposed, as shown in the SEM images above. 

### 3.3. The Density and Diameter of CPFs and ACPFs Analysis

The diameter and density of the fibers affect their properties and applications. From Table 1, it can be seen that the diameter of CPFs was 450 ± 48 μm and the diameter of ACPFs was 350 ± 62 μm. The results showed that alkali treatment can remove impurities from the fiber surface, which made the diameter of the fibers decrease. The density of the CPFs was 0.80 g/cm^−3^ and the density of the ACPFs was 0.92 g/cm^−3^. The results indicated that the fibers had a low density and can be used to make lightweight composites.

### 3.4. Tensile Properties of CPFs and ACPFs

The tensile properties of CPFs and ACPFs are shown in Table 1. CPFs exhibited tensile strength and tensile modulus of 228.70 MPa and 4103.25 MPa. As a result of the partial removal of amorphous contents and the enrichment of cellulose, ACPFs exhibited superior mechanical properties to CPFs. After alkali treatment, the tensile strength of the ACPFs (355.77 MPa) was superior to the 240 MPa value reported by Arunavathi et al. of jute fibers [42]. The tensile modulus of ACPFs was 5212.36 MPa, which was superior to the 3570.00 MPa value reported by Kithiia, Munyasi, and Mutuli of sisal fibers and 4010.00 MPa value reported by Akintayo et al. [43,44]. 

### 3.5. Interfacial Shear Strength

The interfacial shear strength between fibers and PLA is depicted in Table 2. Droplet test results showed that the interfacial shear strength of ACPFs reached 14.06 MPa, which was 130.9% higher than that of raw fibers. This may be attributed to the effect of the alkali treatment, which diminished the lignin and hemicellulose content leading to PLA bonding well with the fibers [45,46]. Holes on the surfaces increased the mechanical interlock points between the fibers and matrix, which greatly improved the interfacial strength [47]. Good interfacial bonding in composites was conducive to the load transfer between the components [48].

### 3.6. Thermogravimetric Analysis

When plant fibers are used in the preparation of composites, they usually undergo several high–temperature processes during the forming process, so it is crucial to determine the thermal degradation behavior of plant fibers [49]. Figure 6 shows the TG and DTG curves of CPFs and ACPFs. It shows that at lower temperatures (50–150 °C), the fiber quality decreased slightly due to the evaporation of water. 

At higher temperatures (200–400 °C), up to 70% or more of fiber mass may be lost due to the decomposition of hemicellulose, cellulose, and lignin. As can be seen from Figure 6a, there were multiple peaks due to surface impurities and some components not being removed. The CPFs had a thermal decomposition peak at 267.8 °C, which was caused by the decomposition of hemicellulose. Hemicellulose was the first chemical component to decompose due to its amorphous structure in the fiber, and its decomposition temperature was usually in the range of 200–300 °C [50].

After alkali treatment, it was found that the decomposition peak became less obvious near 200–300 °C. Because some small molecule impurities such as hemicellulose were dissolved by alkali treatment. Cellulose was formed by microfibrils, which were relatively more stable compared to hemicellulose. It started to degrade when hemicellulose was completely decomposed. Lignin was the last component to decompose under high–temperature conditions. Lignin provided rigid support for plant fibers and imparted higher thermal stability to fibers. The decomposition temperature of cellulose and lignin in plant fibers ranges from approximately 300 to 500 °C [51]. There was a maximum thermal decomposition peak between 300 and 400 °C for the fibers before and after the alkali treatment, where the thermal decomposition of the cellulose and lignin fractions was observed. However, it can be seen that after 2 h of alkali treatment, the maximum thermal decomposition rate temperature was 341.9 °C, which was higher than the 331.4 °C value of CPFs. Since the alkali treatment removed some molecules that were less thermally stable, this led to an improvement in the thermal stability of the fibers.

### 3.7. SEM Analysis of Composites

SEM reflects a clear picture of the interfacial adhesion, fibers distribution, and failure modes. For this purpose, SEM of PLA/ACPF composites with different fiber loadings of 10, 20, 30, 40, and 50 wt.% was carried out and the images are presented in Figure 7. It was seen that the matrix packed the ACPFs closely, and there was good interfacial bonding between them. Figure 7a shows that the fibers were not uniformly distributed in the matrix when the fiber content was low. The cross–section was flat, and only a few fibers were pulled out or broken. With the gradual increase of fiber loadings, as shown in Figure 7c–e, the fibers were randomly and irregularly distributed in the matrix. There were more fibers pulled out or broken. Some fibers parallel to the fracture surface were torn or pulled out laterally. The cross–section became uneven. Figure 7e depicts a lower void content within the composites. Some of the fibers were connected to each other. Additionally, Figure 7e shows that when the fibers were pulled out, the surface was partially coated with PLA matrix, indicating good interfacial adhesion between them. A stress transfer network was formed within the composites, which facilitated the transfer of loads.

### 3.8. Mechanical Properties of Composites

Strength, stiffness, and toughness are the most important performance for structural composites. Figure 8 highlights the relationship between modulus and fiber loadings. Fiber loadings from 10 wt.% to 50 wt.% demonstrated an up–trend in modulus. When the fiber loadings were 50 wt.%, the tensile modulus and the flexural modulus of composites reached 2793.90 MPa and 6959.70 MPa, which were over 50% upgrades relative to the value of pure PLA. The tensile modulus was higher than the 1395.73 MPa value of hemp/PLA composites reported by Wang et al. [52]. The flexural modulus was high than the 4400 MPa value of kenaf mats/PLA composites reported by Manral and Bajpai [53]. The result could be traced to good interfacial adhesion between the fibers and matrix. The ACPFs and the matrix participated in bearing load together and resisted the deformation of composites when the composites were applied by external stress.

Figure 9a highlights the relationship between strength and fiber loadings. The flexural strength changed little with composition. When the fiber loadings were 50 wt.%, the flexural strength of composites reached 111.73 MPa, which was close to that of pure PLA. The flexural strength of composites was high than the 57.00 MPa value of date palm fiber/PLA composites reported by Awad et al. [54]. Fiber loadings from 10 wt.% to 50 wt.% demonstrated an up–trend in tensile strength. When the content increased to 50 wt.%, the tensile strength of composites reached 63.54 MPa, which was higher than that of pure PLA. The tensile strength was also higher than the 50.82 MPa value of nettle/PLA composites and 46 MPa value of ramie/PLA composites reported by Bogard et al. [55]. At low fiber loadings, the load–carrying capacity of the fibers was low. This tended to cause stress concentration and reduce mechanical strength. As the fiber loadings increased, the number of fibers larger than the critical length of the fiber increased, which will further bear the role of external load and avoid the rapid growth of cracks, thus improving the mechanical properties of the composites. As the fiber loadings increased, the fibers in the composite can form a stress transfer network, and the stress transfer efficiency between fibers was improved, thus improving the tensile strength of the composites.

As shown in Figure 10, the elongation at break changed from 6.4% of the pure PLA to 2.6% of the composites and did not change much with composition. When the fiber loadings were low, the stress accumulated on the resin matrix and caused high elongation. As the fiber loadings increased, effective stress transfer between the fibers and resin led to less elongation.

Figure 11 highlights the relationship between impact strength and fiber loadings. Fiber loadings from 10 wt.% to 50 wt.% demonstrated an up–trend in impact strength. The impact strength of PLA/ACPF composites with 50 wt.% fiber content reached 8.2 kJ/m^2^, more than two folds of pure PLA. It was higher than the 1.7 kJ/m^2^ value for PLA/sisal fibers and 2.25 kJ/m^2^ value for PLA/coir fibers reported by Duan et al. [37]. Because the bonding of the fibers with the matrix was good. The fibers absorbed more energy when they were pulled out. The SEM analysis showed that the fracture of the matrix was accompanied by the tearing and fracture of the fibers, which also absorbed much energy [56,57,58]. With the increase in fiber loadings, more fibers participated in the impact load.

## 4. Conclusions

This study presents the characterization of a new fiber named coconut petiole fiber. It can be used as reinforcement in polymer composites. Coconut petiole fibers were extracted and their unique helical and porous structure was characterized by scanning electron microscopy after alkali treatment. The low density (0.92 g/cm^3^) and sufficient tensile strength (355.77 MPa) of ACPFs enhanced the specific strength of polymer composites when used as reinforcement. Micro–sized holes on ACPFs ensured better bonding characteristics between the fibers and matrix. The tensile, flexural, and impact properties of the composites were highly influenced by the fiber loadings. When the fiber loadings reached 50 wt.%, the tensile modulus and the flexural modulus of composites were 2793.90 MPa and 6959.70 MPa, which were over 50% higher than that of pure PLA. The impact strength was 8.2 kJ/m^2^, more than two folds of pure PLA. Hence it was concluded that PLA/ACPFs composites with 50 wt.% fiber loadings showed higher mechanical properties and could be used in automotive and construction fields.

## Figures and Tables

**Figure 1 polymers-15-02200-f001:**
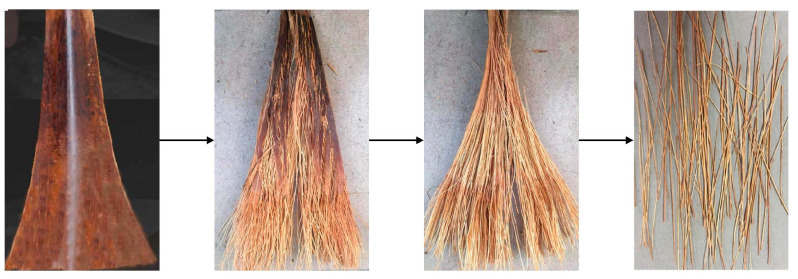
The extraction process of CPFs.

**Figure 2 polymers-15-02200-f002:**
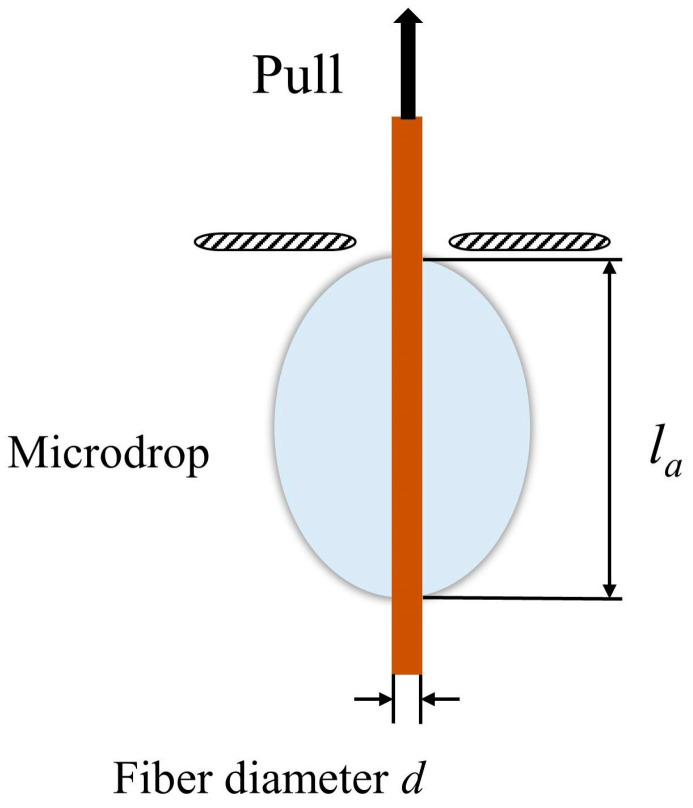
Schematic of microdroplet extraction.

**Figure 3 polymers-15-02200-f003:**
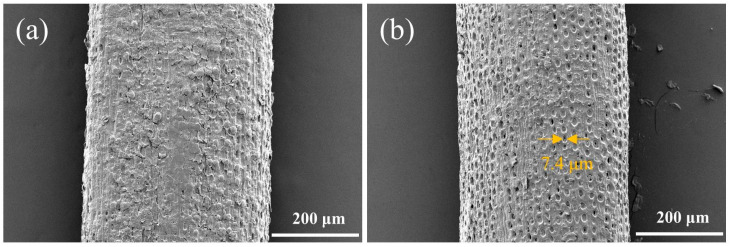
Surface morphology of (**a**) CPFs and (**b**) ACPFs.

**Figure 4 polymers-15-02200-f004:**
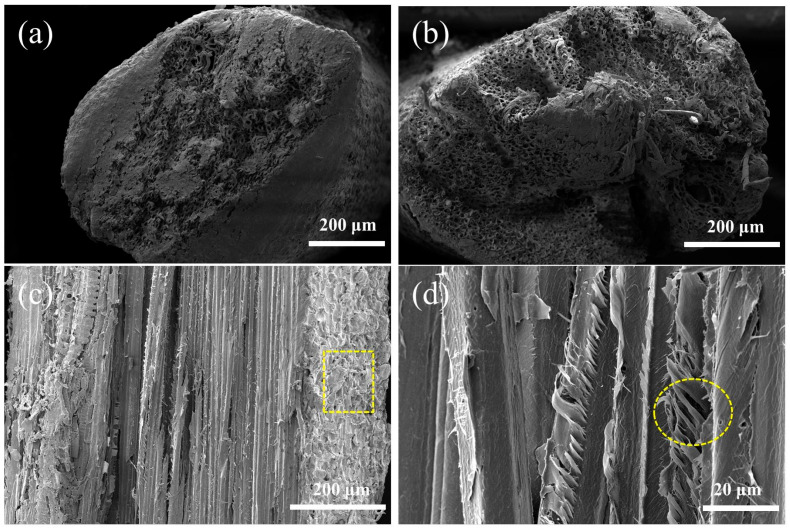
Cross–section SEM images of (**a**) CPFs and (**b**) ACPFs, and longitudinal section SEM images of (**c**,**d**) CPFs.

**Figure 5 polymers-15-02200-f005:**
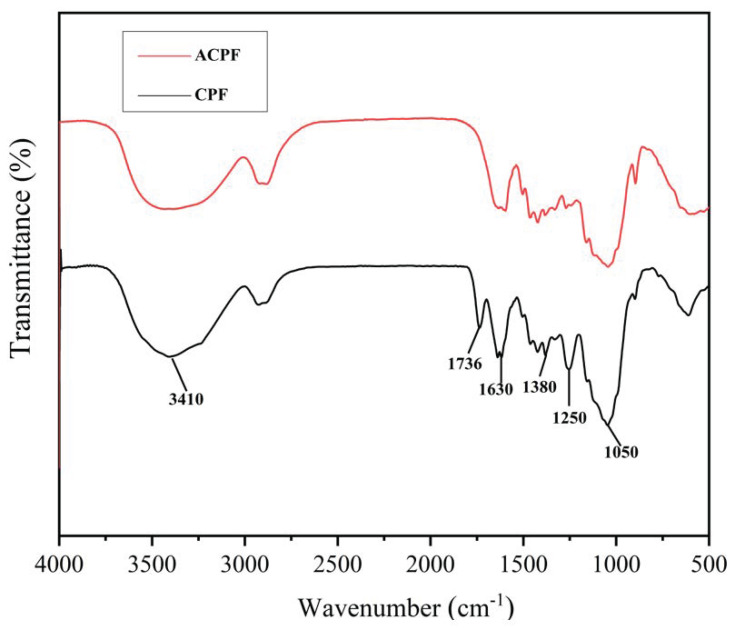
FT–IR spectrum of CPFs and ACPFs.

**Figure 6 polymers-15-02200-f006:**
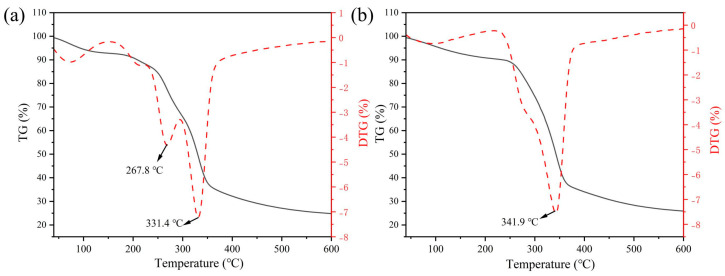
TG and DTG curves of (**a**) CPFs and (**b**) ACPFs.

**Figure 7 polymers-15-02200-f007:**
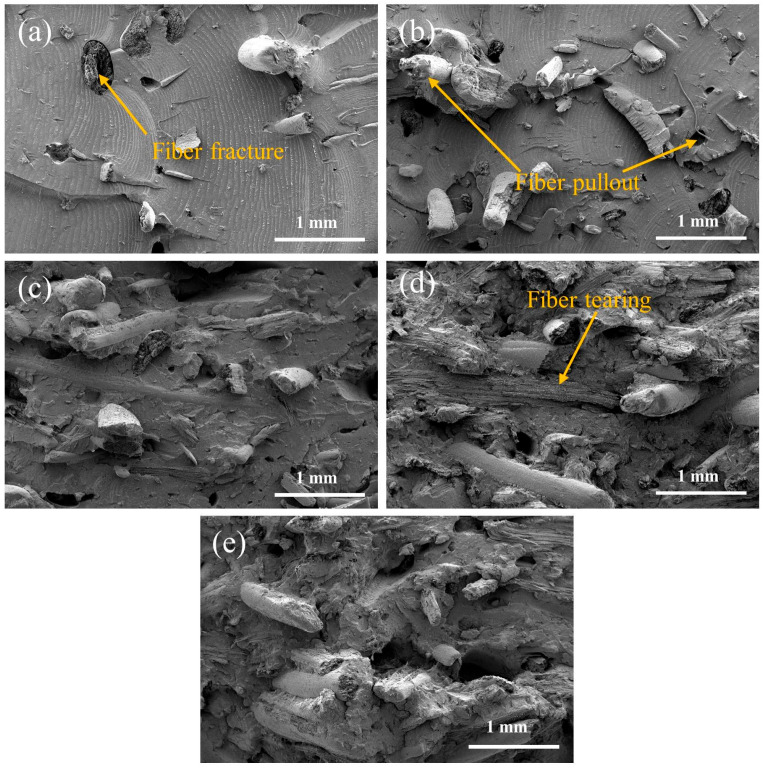
The SEM images of PLA/ACPF composites with different fiber loadings: (**a**) PLA/10%ACPF, (**b**) PLA/20%ACPF, (**c**) PLA/30%ACPF, (**d**) PLA/40%ACPF, (**e**) PLA/50%ACPF.

**Figure 8 polymers-15-02200-f008:**
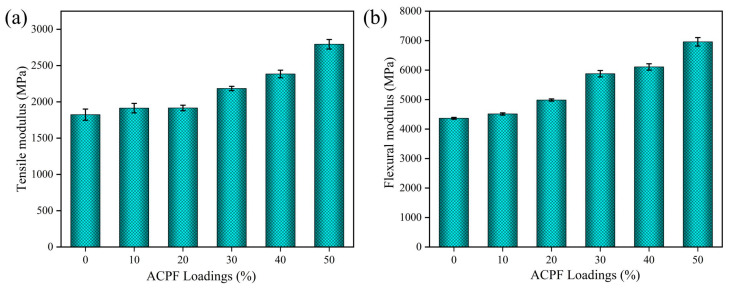
The (**a**) tensile modulus and (**b**) flexural modulus of PLA/ACPF composites with different fiber loadings.

**Figure 9 polymers-15-02200-f009:**
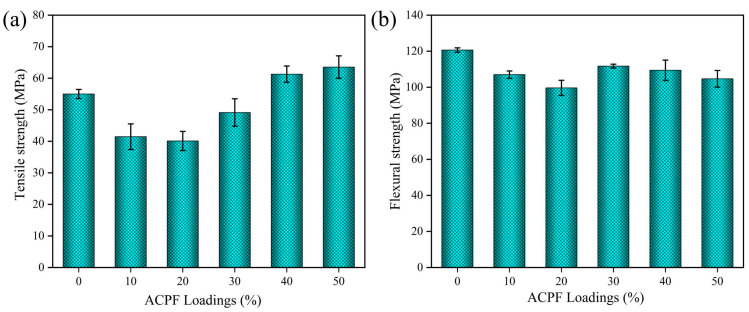
The ultimate strength of PLA/ACPF composites with different fiber loadings: (**a**) tensile strength and (**b**) flexural strength.

**Figure 10 polymers-15-02200-f010:**
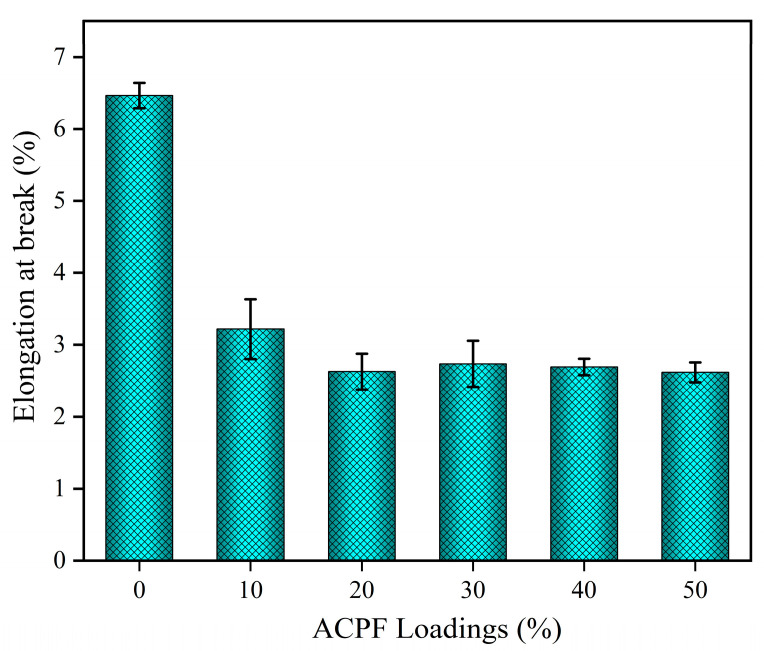
The elongation at break of PLA/ACPF composites with different fiber loadings.

**Figure 11 polymers-15-02200-f011:**
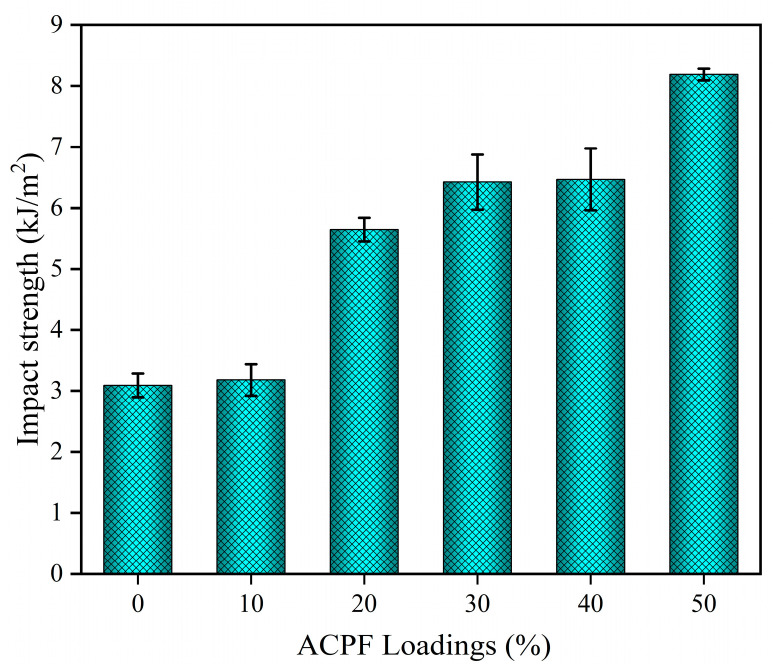
The impact properties of PLA/ACPF composites with different fiber loadings.

**Table 1 polymers-15-02200-t001:** Properties of CPFs and ACPFs.

Fiber	Fiber Diameter (μm)	Fiber Density (g/cm^−3^)	Tensile Strength (MPa)	Tensile Modulus (MPa)
CPF	450 ± 48	0.80	228.70 ± 50.32	4103.25 ± 120.35
ACPF	350 ± 42	0.92	355.77 ± 25.69	5212.36 ± 100.75

**Table 2 polymers-15-02200-t002:** Interfacial shear strength between the fibers and PLA.

Fiber	Interfacial Shear Strength (MPa)
CPF	6.09 ± 0.21
ACPF	14.06 ± 0.32

## Data Availability

The data presented in this study are available on request from the corresponding author. The data are not publicly available due to privacy restrictions.

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
