# Peer review of "The Unique Morphology of Coconut Petiole Fibers Facilitates the Fabrication of Plant Composites with High Impact Performance"

_polymers, 2023, doi:10.3390/polym15092200_

Round 1

Reviewer 1 Report

The manuscript is good and can be accepted for publication upon incorporating the following comments.  Thus revision is suggested at this stage, and comments are as follows.

1.       The introduction should have a detailed emphasis on how climatic conditions, soil, and part of the plant decide the strength of the fibers. Also, give some information on how chemical treatments enhance this strength.

2.       Generally, 5 wt.% of NaOH seems to be beneficial, and many researchers have proved it. Then why specifically 10 wt.% that also 2 hours of treatment were chosen? Give literature evidence to support your claim.

3.       Chemical analysis plays a vital role in discussing strength properties. It is mandatory to do a chemical analysis giving cellulose, hemicellulose, lignin, moisture, and wax for the untreated and chemically treated fibers to support your claim.

4.       How many samples were tested for the fiber tensile test, and what is the allowable error?

5.       The discussion about the cross-section of fiber should be enhanced with more literature correlations; refer to the below article for discussion.

https://doi.org/10.1080/15440478.2019.1612308

6.       Is the strength reported in the composite, ultimate strength, then state it is ultimate tensile and flexural strength?

7.       How the density of the fibers was measured needs to be given in the methods sections

8.       What is the diameter of the treated and untreated fiber? It should be reported.

Author Response

Dear Reviewers:

Thank you very much for your letter and the reviewers’ pertinent comments concerning our manuscript entitled “Unique morphology of coconut petiole fibers facilitates the fabrication of plant composites with high impact performance” (Manuscript ID: polymers-2335963). These comments are all valuable and very helpful for revising and improving our manuscript. We have studied the comments carefully and have made some revision which marked in red in the revised manuscript. Our point-by-point responses to the editor’ and reviewers’ comments are as follow:

Reviewers’ comment & Responses:

The manuscript is good and can be accepted for publication upon incorporating the following comments. Thus revision is suggested at this stage, and comments are as follows.

  1. The introduction should have a detailed emphasis on how climatic conditions, soil, and part of the plant decide the strength of the fibers. Also, give some information on how chemical treatments enhance this strength.

Responses: Thank you very much for your precious suggestion. We added the needed information in the introduction.

Plant fibers are composed of cellulose, hemicellulose, lignin, waxes, and pectin. The climatic conditions, soil and part of the plant decide the ratio of its constituents, which determines the mechanical properties of the fiber [10,11].    [ Page 1, line 32-34]

Arthanarieswaran et al. reported that alkali treatment can partially remove impurities such as hemicellulose, lignin and pectin from the fiber, and the cellulose content and crystallinity increased, which leads to a significant improvement in the mechanical prop-erties of the fiber [21]. Kapatel reported that the fiber/matrix adhesion of jute/epoxy com-posites was improved by removing impurities from the jute fabric surface after alkali treatment, resulting in the improvement of the mechanical properties of the composites [22].      [ Page 1, line 39-45]

  1. Generally, 5 wt.% of NaOH seems to be beneficial, and many researchers have proved it. Then why specifically 10 wt.% that also 2 hours of treatment were chosen? Give literature evidence to support your claim.

Responses: Thank you very much for your precious suggestion. First, we found that 10 wt.% of NaOH can better remove impurities from the fiber surface and improve the interfacial bond between the fiber and the substrate by reading through the literature. We used 10 wt.% of NaOH to treat coconut petiole fibers and studied the properties of the fibers at different times. We found that the highest mechanical properties and interfacial strength of coconut petiole fibers were obtained at 2h, so we finally chose 10 wt.% of NaOH and 2h.

Literature evidence

Sullins, T.; Pillay, S.; Komus, A.; Ning, H. Hemp fiber reinforced polypropylene composites: The effects of material treatments. Composites Part B: Engineering 2017, 114, 15-22, doi:10.1016/j.compositesb.2017.02.001.

Pickering, K.L.; Beckermann, G.; Alam, S.; Foreman, N.J. Optimising industrial hemp fibre for composites. Composites Part A: Applied Science and Manufacturing 2007, 38, 461-468, doi:10.1016/j.compositesa.2006.02.020.

  1. Chemical analysis plays a vital role in discussing strength properties. It is mandatory to do a chemical analysis giving cellulose, hemicellulose, lignin, moisture, and wax for the untreated and chemically treated fibers to support your claim.

Responses: Thank you very much for your precious suggestion. On the one hand, we demonstrated through infrared tests that the alkali treatment removed hemicellulose, lignin and pectin, while the cellulose content was enhanced, thus improving the tensile properties of the fibers and the interfacial adhesion of the fibers and the PLA. On the other hand, the effect of alkali treatment on the chemical composition of fibers was proved and reported by many researchers, and we presented it in the introduction by citing the literatures. We focused more on the mechanical properties of the alkali-treated coconut petiole fibers and their reinforcing effect on the composites, and we will do a chemical analysis of the components when we conduct subsequent fiber composition studies.

  1. How many samples were tested for the fiber tensile test, and what is the allowable error?

Responses: Thank you very much for your precious suggestion. Five samples were measured for each test and the average values were recorded. We added the allowable error in Table 1.     

Table 1. Properties of CPFs and ACPFs.

Fiber

Fiber diameter

(μm)

Fiber density

(g/cm-3)

Tensile strength MPa

Tensile modulus (MPa)

CPF

450±48

0.80

228.70±50.32

4103.25±120.35

ACPF

350±42

0.92

355.77±25.69

5212.36±100.75

[ Page 7, line 230]

  1. The discussion about the cross-section of fiber should be enhanced with more literature correlations; refer to the below article for discussion.

https://doi.org/10.1080/15440478.2019.1612308

Responses: Thank you very much for your precious suggestion. We have discussed the cross-section of fiber referring to the above article.

Figure 4 shows the sectional morphology of CPFs and ACPFs. The fibers were com-posed of a series of microfiber bundles as reported in Figure 4a-4b. This structure of CPFs and ACPFs had similarities to that of sisal fibers in work completed by Lv et al. [32]. Compared to the CPFs, the binding components on the surface of the ACPFs as well as the binding components between the fiber bundles were removed and there was a slight col-lapse of the fiber bundles. The cross-section of the ACPFs was rough, and many grooves appeared, which was typified in the investigation carried out by Vijay et al. [40]. The lon-gitudinal section morphology of the fibers might offer further information about the structure of CPFs. The microfibers presented an obvious spiral-pattern structure in Figure 4c and 4d. This special helical structure might have a significant impact on the mechani-cal properties of the fibers. When the fibers were pulled in tension, these microfibers might uncoil like springs with bending and twisting, thus giving the fibers a certain degree of tensile toughness.    [ Page 5, line 179-191]

  1. Is the strength reported in the composite, ultimate strength, then state it is ultimate tensile and flexural strength?

Responses: Thank you very much for your precious suggestion. The tensile strength and flexural strength were reported in Page 10, line 304-319.

Figure 9a highlights the relationship between strength and fiber loadings. The flexur-al strength changed little with composition. When the fiber loadings were 50 wt. %, the flexural strength of composites reached 111.73 MPa, which was close to that of pure PLA. The flexural strength of composites was high than 57.00 MPa value of date palm fi-ber/PLA composites reported by Awad et al. [54]. Fiber loadings from 10 wt. % to 50 wt. % demonstrated an up-trend in tensile strength. When the content increased to 50 wt. %, the tensile strength of composites reached 63.54 MPa, which was higher than that of pure PLA. The tensile strength was also higher than 50.82 MPa value of Nettle/PLA composites and 46 MPa value of Ramie/PLA composites reported by Bogard et al. [55]. At low fiber loadings, the load-carrying capacity of the fibers was low. This tended to cause stress concentration and reduce the mechanical strength. As the fiber loadings increased, the number of fibers larger than the critical length of the fiber increased, which will further bear the role of external load and avoid the rapid growth of cracks, thus improving the mechanical properties of the composites. As the fiber loadings increased, the fibers in the composite can form a stress transfer network, and the stress transfer efficiency between fi-bers was improved, thus improving the tensile strength of the composites.                    [ Page 10, line 303-318]

  1. How the density of the fibers was measured needs to be given in the methods sections?

Responses: Thank you very much for your precious suggestion. We gave the method of measuring the density of the fibers in the methods sections.

The pycnometer was used to test the density of fibers, mainly using the mass difference technique. The reagent used for the density test was liquid acetone with a density of 0.7845g/cm-3. First, the fibers were put into the pycnometer, and the masses of the pycnometer before and after the placement were recorded as m1 and m2, respectively. A certain mass of acetone (m3) was added to the pycnometer. The final mass m4 was recorded and the density of the fibers could be calculated by the following equation:

(1)

[ Page 3, line 118-124]

  1. What is the diameter of the treated and untreated fiber? It should be reported.

Responses: Thank you very much for your precious suggestion. We added the diameter of the treated and untreated fiber in Table 1.     

Table 1. Properties of CPFs and ACPFs.

Fiber

Fiber diameter

(μm)

Fiber density

(g/cm-3)

Tensile strength MPa

Tensile modulus (MPa)

CPF

450±48

0.80

228.70±50.32

4103.25±120.35

ACPF

350±42

0.92

355.77±25.69

5212.36±100.75

[ Page 7, line 230]

Reviewer 2 Report

The authors present the morphological, physical, mechanical, and interfacial properties of alkali-treated coconut flower fibers (ACPFs) and the effects of fiber loading on the mechanical properties of ACPFs/PLA composites. The composites achieved high impact strength and flexural modulus, which were 2.5 times and 1.5 times higher than those of pure PLA, respectively.

The work is important because it addresses the current issue of studying new natural composites. The research on ACPFs/PLA composites is new and has not been published before. The manuscript is well written and the conclusions are supported by the research, so I can recommend the paper for publication.

What is missing in the paper is a comparison of the properties of ACPFs/PLA composites with other natural composites, for example, with the results of their recently published work: Liang, Zhifang, et al. "Preparation of long sisal fiber-reinforced polylactic acid biocomposites with highly improved mechanical performance." Polymers 13.7 (2021): 1124. In this sense, the article can also be improved by adding a review article, e.g., Zhao, Xianhui, et al. "Recycling of natural fiber composites: Challenges and opportunities" Resources, Conservation, and Recycling 177 (2022): 105962.

The labels and text on most images are not legible due to low resolution. This has yet to be corrected.

Author Response

Dear Reviewers:

Thank you very much for your letter and the reviewers’ pertinent comments concerning our manuscript entitled “Unique morphology of coconut petiole fibers facilitates the fabrication of plant composites with high impact performance” (Manuscript ID: polymers-2335963). These comments are all valuable and very helpful for revising and improving our manuscript. We have studied the comments carefully and have made some revision which marked in red in the revised manuscript. Our point-by-point responses to the editor’ and reviewers’ comments are as follow:

The authors present the morphological, physical, mechanical, and interfacial properties of alkali-treated coconut flower fibers (ACPFs) and the effects of fiber loading on the mechanical properties of ACPFs/PLA composites. The composites achieved high impact strength and flexural modulus, which were 2.5 times and 1.5 times higher than those of pure PLA, respectively.

The work is important because it addresses the current issue of studying new natural composites. The research on ACPFs/PLA composites is new and has not been published before. The manuscript is well written and the conclusions are supported by the research, so I can recommend the paper for publication.

What is missing in the paper is a comparison of the properties of ACPFs/PLA composites with other natural composites, for example, with the results of their recently published work: Liang, Zhifang, et al. "Preparation of long sisal fiber-reinforced polylactic acid biocomposites with highly improved mechanical performance." Polymers 13.7 (2021): 1124. In this sense, the article can also be improved by adding a review article, e.g., Zhao, Xianhui, et al. "Recycling of natural fiber composites: Challenges and opportunities" Resources, Conservation, and Recycling 177 (2022): 105962.

Responses: Thank you very much for your precious suggestion. We have added the comparison of the properties of ACPFs/PLA composites with other natural composites.

(1) Strength, stiffness, and toughness are the most important performance for structural composites. Figure 8 highlights the relationship between modulus and fiber loadings. Fiber loadings from 10 wt. % to 50 wt. % demonstrated an up-trend in modulus. When the fiber loadings were 50 wt. %, the tensile modulus and the flexural modulus of composites reached 2793.90 MPa and 6959.70 MPa, which were over 50% upgrades relative to the value of pure PLA. The tensile modulus was higher than 1395.73 MPa value of hemp/PLA composites reported by Wang et al. [52]. The flexural modulus was high than 4400 MPa value of kenaf mats/PLA composites reported by Manral and Bajpai [53]. The result could be traced to good interfacial adhesion between the fibers and matrix. The ACPFs and the matrix participated in bearing load together and resisted the deformation of composites when the composites were applied by external stress.

[ Page 9, line 289-299]

Wang, J.; Bai, J.; Hua, H.; Tang, B.; Bai, W.; Wang, X. Characterization and Scalable Production of Industrial Hemp Fiber Filled PLA bio-composites. Journal of Natural Fibers 2022, 19, 13426-13437, doi:10.1080/15440478.2022.2095549.

Manral, A.; Bajpai, P.K. Effect of non-acidic chemical treatment of kenaf fiber on physico mechanical properties of PLA based composites. Journal of Natural Fibers 2022, 19, 5709-5727, doi:10.1080/15440478.2021.1889435.

(2) Figure 9a highlights the relationship between strength and fiber loadings. The flexur-al strength changed little with composition. When the fiber loadings were 50 wt. %, the flexural strength of composites reached 111.73 MPa, which was close to that of pure PLA. The flexural strength of composites was high than 57.00 MPa value of date palm fi-ber/PLA composites reported by Awad et al. [54]. Fiber loadings from 10 wt. % to 50 wt. % demonstrated an up-trend in tensile strength. When the content increased to 50 wt. %, the tensile strength of composites reached 63.54 MPa, which was higher than that of pure PLA. The tensile strength was also higher than 50.82 MPa value of Nettle/PLA composites and 46 MPa value of Ramie/PLA composites reported by Bogard et al. [55]. At low fiber loadings, the load-carrying capacity of the fibers was low. This tended to cause stress concentration and reduce the mechanical strength. As the fiber loadings increased, the number of fibers larger than the critical length of the fiber increased, which will further bear the role of external load and avoid the rapid growth of cracks, thus improving the mechanical properties of the composites. As the fiber loadings increased, the fibers in the composite can form a stress transfer network, and the stress transfer efficiency between fi-bers was improved, thus improving the tensile strength of the composites.    [ Page 10, line 303-318]

Awad, S.; Hamouda, T.; Midani, M.; Katsou, E.; Fan, M. Polylactic Acid (PLA) Reinforced with Date Palm Sheath Fiber Bio-Composites: Evaluation of Fiber Density, Geometry, and Content on the Physical and Mechanical Properties. Journal of Natural Fibers 2023, 20, 2143979, doi:10.1080/15440478.2022.2143979.

Bogard, F.; Bach, T.; Abbes, B.; Bliard, C.; Maalouf, C.; Bogard, V.; Beaumont, F.; Polidori, G. A comparative review of Nettle and Ramie fiber and their use in biocomposites, particularly with a PLA matrix. Journal of Natural Fibers 2022, 19, 8205-8229, doi:10.1080/15440478.2021.1961341.

(3) Figure 11 highlights the relationship between impact strength and fiber loadings. Fi-ber loadings from 10 wt. % to 50 wt. % demonstrated an up-trend in impact strength. The impact strength of PLA/ACPF composites with 50 wt. % fiber content reached 8.2 kJ/m2, more than two folds of pure PLA. It was higher than the 1.7 kJ/m2 value for PLA/sisal fi-bers and 2.25 kJ/m2 value for PLA/coir fibers reported by Duan et al. [37]. Because the bonding of the fibers with the matrix was good. The fibers absorbed more energy when they were pulled out. The SEM analysis showed that the fracture of the matrix was ac-companied by the tearing and fracture of the fibers, which also absorbed much energy [56-58]. With the increase of fiber loadings, more fibers participated in the impact load.                         [ Page 11, line 329-337]

Duan, J.P.; Wu, H.W.; Fu, W.C.; Hao, M.Y. Mechanical properties of hybrid sisal/coir fibers reinforced polylactide biocomposites. Polymer Composites 2018, 39, E188-E199, doi:10.1002/pc.24489.

The labels and text on most images are not legible due to low resolution. This has yet to be corrected.

Responses: Thank you very much for your precious suggestion. We have recreated the images to ensure the labels and text on images were legible.
